# Geometry-Dependent Efficiency of Dean-Flow Affected Lateral Particle Focusing and Separation in Periodically Inhomogeneous Microfluidic Channels

**DOI:** 10.3390/s22093474

**Published:** 2022-05-03

**Authors:** Anita Bányai, Eszter Leelőssyné Tóth, Máté Varga, Péter Fürjes

**Affiliations:** 1Microsystems Laboratory, Centre for Energy Research, Institute of Technical Physics and Materials Science, ELKH, Konkoly Thege Miklós Str. 29-33, H-1121 Budapest, Hungary; leelossyne.toth.eszter@ek-cer.hu (E.L.T.); furjes.peter@ek-cer.hu (P.F.); 277 Elektronika Ltd., Fehérvári Str. 98, XI., H-1111 Budapest, Hungary; mvarga@e77.hu; 3Doctoral School on Materials Sciences and Technologies, Óbuda University, Bécsi Str. 96/B, H-1034 Budapest, Hungary

**Keywords:** cell manipulation, dean flow, hydrodynamic lift, microfluidics, computational fluid dynamics

## Abstract

In this study, inertial focusing phenomenon was investigated, which can be used as a passive method for sample preparation and target manipulation in case of particulate suspensions. Asymmetric channel geometry was designed to apply additional inertial forces besides lift forces to promote laterally ordered particles to achieve sheathless focusing or size-dependent sorting. The evolving hydrodynamic forces were tailored with altered channel parameters (width and height), and different flow rates, to get a better understanding of smaller beads’ lateral migration. Fluorescent beads (with the diameter of 4.8 µm and 15.8 µm) were used to distinguish the focusing position in continuous flow, and experimental results were compared to in silico models for particle movement prediction, made in COMSOL Multiphysics. The focusing behaviour of the applied microfluidic system was mainly characterised for particle size in the range close to blood cells and bacteria.

## 1. Introduction

The application of portable medical devices such as Lab-on-chip (LOC)-based Point-of-Care (POC) diagnostic systems are able to accelerate diagnostic decisions regarding certain diseases, infections, with the involvement of biological markers. These emerging applications can provide reliable diagnosis in a few minutes, supporting on- time decision of further medical treatments. Rising attention focuses on the preparation, analysis and treatment of cell populations or single cells in special microsystems. There is an indisputable need for these tools, therefore intensive research has been launched in recent years to better understand the possibility and limitations of these miniaturized systems and extend their opportunities regarding cell-scale manipulations.

Sample preparation and target isolation are essential tasks, which can be achieved by the hydrodynamic forces arising from the inertia of the fluid itself. Inertial focusing was discovered in 1961 [1], and even today, intensively researched phenomenon be used for target enrichment, separation, focusing, sample filtration as a result, without any specific particle labelling or other preparation.

The passive separation method takes precedence over active separation methods—like dielectrophoretic [2] acoustophoretic [3], magnetic [4], optic [5], or even thermophoretic separation [6]—because no external resource is required to manipulate the fluid contents. Its effective function depends mostly on channel geometry and inherent hydrodynamic forces, which makes the analytical platform uncomplicated and more portable, accordingly. The parameterization of microfluidic devices, however, requires expertise and special attention, considering target cell shape [7], size [8], elasticity, or deformability [9]. The phenomenon can be used in connection with food industry or medical care to separate pathogen bacteria in continuous flow with micron-scale resolution [10] as well as to isolate rare cells—such as circulating tumor cells [11] or bacteria from whole blood with the help of sheath flow in a spiral channel [12]. Not just the minimal size of the target, but the biological medium itself makes the separation especially difficult. Microchannel geometry design depends on whether incompressible Newtonian or non-Newtonian, viscoelastic fluid is maintained in separation.

In case of viscoelastic fluids—such as poly(ethylene oxide) (PEO) [13]—the characteristic features are the flatter velocity profile, even changing viscosity with increasing shear stress. Beyond the general inertial forces, further phenomenon, such as the presence of elastic and viscous forces, and their ratio to inertial forces need to be considered. Due to the elastic forces, the shear thinning, and particle deformability and particle interactions, more diverse equilibrium position patterns can be observed in the flow cross-section compared to Newtonian flows. Accordingly, the particle behavior is also more difficult to predict by computational fluid dynamics (CFD) models [14].

Present work studies the realistic approach of Newtonian fluids and the challenge of the focusing of smaller particles utilising solely hydrodynamic principles in microchannel having various geometry. The performance of an asymmetric curvilinear system were investigated—described by Di Carlo et al. [15]—to localize the focusing positions in the channel for different particle sizes. Nivedita et al. [16] made a comprehensive research regarding the development and formation of Dean vortices in the channel cross-section considering the change of wall aspect ratio, geometry and volumetric flow rate. Critical Dean number were determined, as the limit of formation secondary vortices. The lateral focusing effect can be supported by the Dean flow evolving secondary transversal flow components perpendicular to the primary flow. This effect can be visualized as two counter rotating, recirculating vortices in the channel cross-section. In straight channels, according to the parabolic flow profile the highest flow rate is measurable at the centerline of the channel. In contrast, in a meandering channel, this velocity maximum is shifted in the direction of the larger radius of the bend, and the velocity-field-induced pressure gradient triggers a specific transverse flow component. We are able to utilize this phenomenon to decrease the spreading and tune the position of the focus points within the channel.

In our study, the position and extent of focused region were investigated using polystyrene fluorescent beads with different bead diameters (4.8 µm and 15.8 µm) in a curved channel at different flow rates (0.5–6 µL/s). The microchannel was parameterized by its height and width of the critical (strictured) cross section. The extent/effectivity of focusing was examined at the end of the channel in the broad section. Here, the fluorescent intensity of the fluorescent beads was captured, to visualize their spatial distribution, and the lateral focusing position in channel cross-section were determined by the intensity maximum. For better comprehension of size-dependent particle behaviors computational fluid dynamics model also was made in COMSOL Multiphysics software, and the results were compared to the experimental results. The aim of this study was to improve the possibility and focusing efficiency of the smaller beads close or under the typical cell sizes (RBC—6 µm), whether by enlargement the flow rate or by optimization channel geometry.

## 2. Theoretical Background

In the field of passive particulate focusing, several concept, channel geometry were used to tone down the manipulable target size, although the very first observations of inertial based annular particle orientations was made in circle shaped, straight tube by Segre and Silberberg [1]. The tubular pinch effect is a form of inertial migration—induced by the shear-gradient lift force—where rigid, spherical particles migrate to a specific equilibrium position, forms an annulus—at ~0.6 times the tube radius—at low Reynolds numbers in cylindrical Poiseullie flow. The wall-effect-induced lift force [17], in turn, due to asymmetric fluid velocity, and higher pressure at the walls, drives away the particles from the periphery. The balance between inertial lift forces can be further tailored by a third factor, the secondary-flow drag force [18], which can be either evoked at low Reynolds numbers in a curvature-induced secondary turbulent flow—Dean vortex; or can be forced to occur in straight channel at higher Reynolds number [17]. The most important proportional parameters, which characterize the focusing effects [15,19,20], are
Reynolds number of the channel (*Re_c_*);Based on fluid density, viscosity, the maximum velocity of fluid, compared to the hydraulic diameter (*D_h_*);Ratio between the Reynolds number of the particle (*Re_p_*) and particle size (*a*), and channel cross-section (*D_h_*);Magnitude of lift forces (*F_z_*);Particle migration velocity balanced with Stokes drag force (*F_s_*);Non-dimensional Dean number (*De*) characterizing the secondary flow.

(1)$$De=Re_{c\;}\sqrt{\frac{D_{h}}{2R}},$$
where the radius of the curvature is noted as *R*


7.Magnitude of rotational flow velocity and the secondary-flow-induced Dean drag force (*F_D_*);8.A lift coefficient (*f_c_*) is considered, which depends on particle position in flow, and the channel Reynolds number.


It was demonstrated by Di Carlo et al. [15] that curvature ratio of channel geometry (δ), channel Reynolds number (*Re_c_*), particle diameter (*a*), and hydraulic diameter (*D_h_*), strictly parametrize the evoked force balance in the flow (*n* < 0):(2)$$\frac{F_{z}}{F_{D}}\sim \frac{1}{\delta }\;{\left(\frac{a}{D_{h}}\right)}^{3}Re_{c}^{n},$$

The effect of a weak Dean flow can help focusing, but in case of the Dean drag force becomes dominant, that prevents the process. They suggested adequate geometric relations for the design: for choosing channel length, the inertial migration speed of the smaller particles needs to be considered, and they also determined the *a*/*D_h_* ratio to be above 0.07, enabling the focusing effect.

By choosing the applicable channel geometry for particle focusing, such as the above-mentioned asymmetric curved serpentine [15], curved [19,21], spiral channels [22,23] or a periodic series of channel contractions and expansions [17,24], the particle size contributes to the theoretical separation scaling to a varying degrees [18] and affect the inertial migration speed of the particle. In a straight square cross-section (at aspect ratio, *AR = h/w = 1*), the number of equilibrium positions can be reduced to 4 points at the channel faces as demonstrated in Figure 1 schematically. In case of straight rectangular shaped microchannel this number can be reduced to 2 equilibrium positions, but not further. The chosen wall-size aspect ratio can significantly affect the focusing pattern, because particles tend to migrate to the wall, with the longer channel face. In curved geometries the channel height is usually lower, than the channel width, and particles tend to shift to the inside edge of the channel from the longitudinal [21,25]. In asymmetric curved channels—due to the counter rotation Dean flow perpendicular to the primary flow direction in this case—the number of focusing positions can even be reduced to 1 [15].

In summary, the particle size [26], the channel shape (straight or curved), the geometry [27,28], as channel wall aspect ratio [15,25], and the flow Reynolds number [29] influence the focusing positions of the particles in the cross-section of a microfluidic channel. The challenge is to obtain enhanced particle concentration and focusing for smaller beads keeping the channel length and fluid velocity manageable.

## 3. Materials and Methods

### 3.1. Fabrication of Microfluidic Test Structures

The polymer microfluidic chips were fabricated by soft lithography by Polydimethylsiloxane (PDMS). SU-8 2025/2050/2100 epoxy based negative photoresists were applied as a moulding replica and patterned by broadband UV photolithographic exposure in Süss MicroTech MA6 mask aligner after spin-coated on silicon wafer and pre-baked at 65 °C and 95 °C by Brewer Science Cee 200CBX spin-bake system. The pattern was developed in Süss MicroTech spray developer and baked at 95 °C on hotplate again. Thicknesses of the moulding replica were set around 25 µm, 50 µm and 100 µm, according to the applied SU-8 types. PDMS pre-polymer (with 1:10 elastomer/curing agent ratio) was poured onto the replica and cured in oven for 90 min at 65 °C. Finally, the PDMS microfluidic chip was sealed to microscope glass side by low temperature bonding after oxygen plasma treatment using 50 W plasma power, 100 kPa chamber pressure and 1400–1900 sccm oxygen flow in Diener Pico plasma etcher.

### 3.2. Design Aspects

The applied geometries contain periodic sequence of asymmetrically curved serpentine channels. The overall channel length is ~35 mm, including 23 curvatures, with a narrower, and a wider curve (see Figure 2). In our case, the channel geometry parameters are varying as described:The widths of the smaller bend: 100/150/200 µm.The sizes of the critical width (W_cr_): 50/100/150 µm, which defines the smallest cross section in the microfluidic systems.The width of the wider curve (300 µm), and the degree of deflection for small and large apex edges (1000 µm) are the same.Each structure was generated in three different heights (H) 25/50/100 µm. The parameter of these structures can be defined later by choosing adequate SU-8 types based on the channel height (H), and the critical width (W_cr_).

The design of photolithographic mask was made in CleWin5 vector graphical software.

### 3.3. Finite Element Modelling of Particle Behaviour in the Microfluidics

A computational fluid dynamics (CFD) simulation was also performed using COMSOL Multiphysics (version 5.3a) to analyse and predict particle movement in the specially designed microchannels. Finite Element Modeling (FEM) is applied to numerically calculate the Navier–Stokes equation considering laminar flow due to the low Reynolds number regime [30]. In this case, laminar inflow boundary was applied to the inlet with varying flow rates (0.5–6 µL/s), zero backpressure with suppressed backflow was used as outlet boundary condition and ‘no slip’ feature was set for the channel walls. The mechanical properties of room temperature water (density: 1000 kg/m^3^, kinematic viscosity: 10^−6^ m^2^/s) were set as material parameters. Maximal cell Reynolds number varied between 0.02–0.7, the volume average cell Reynolds number was between 0.002–0.08 in all cases.

Particle tracing module was used to calculate particle trajectories in the pre-solved velocity field. 5000 spherical particles with two different diameters were released with uniform distribution from the inlet surface. Freeze boundary condition was applied at the outlet and stick boundary was set for the channel walls. Particle properties were set to be in correspondence with fluorescently labeled polystyrene beads applied in experimental validation (density: 1055 kg/m^3^, diameter: 4.8 µm, 15.8 µm). Exploiting structure periodicity 3D geometry was built for the representative fraction (2 waves) of the microfluidic channel, and periodic boundary condition (continuity) was applied at the outlet of the section mapping the particles back to the inlet with their last position to gain data for the full channel length. Tetrahedral mesh was applied for the 3D models with approximately 16,000,000 elements.

### 3.4. Data Processing for Model Verification

Experimental data were obtained by ImageJ [31] as lateral intensity distribution through the microchannel from images recorded by fluorescent microscopy using the adequate bandpass filter sets (see Figure 3(A1,A2)). To compare the experimental and simulated theoretical results the relative lateral distributions of the particles were derived from the data sets as the function of the *y* coordinate (perpendicularly to the flow direction) at a given cross sectional plane.

Relative fluorescent intensity distribution was calculated (see Figure 3(A1,A2)) by transforming the recorded intensities at the widest cross-sectional plane of the periodic microfluidic channel in the last section. According to the minimal height of the microchannels, linear correspondence was supposed between the particle distribution and the fluorescent intensity recorded. To ensure comparability between the experimental and modelled results the line integral of the distribution functions through the channel width were set to be 1, as presented by Equation (3):(3)$$\int_{0}^{w}I_{1}\left(x_{0},\;y\right)\;dy=\int_{0}^{w}I_{2}\left(x_{0},\;y\right)\;dy=1\;$$
where *y* represents the local coordinates in the channel across a given cross-section at *x*_0_ coordinate. Accordingly, *w* denotes the width of the microchannel, *I*_1_ and *I*_2_ represent the intensity functions of the two different particle sets.

Modeling data were collected by exporting particle coordinates and diameters for all timesteps. Periodic particle trajectories were extended to the whole channel length and lateral particle location Poincaré map was created from the particle dataset at the widest *y-z* plane of the microchannel (see Figure 3(B1)). Particle distribution was calculated from the modeling data extending the center coordinates with 1440 points within the particle radius calculating a histogram of these extended points across the channel. Calculated distributions were normalized to ensure the comparability to the optical measurements as demonstrated by Equation (3).

## 4. Results and Discussions

### 4.1. Geometry Dependency of the Lateral Focusing Effect in Low Aspect Ratio Microchannels

The particle focusing efficiencies were characterized by the fluorescent intensity maximum of the different beads, and by their lateral position in the last—23rd—, wide curvature having 300 µm width in each fluidic structure. The focusing process is more efficient in case the intensity function has characteristic sharp peaks indicating the focusing position in the microchannels. Since our goal was to study the focusing behaviour of our microfludic geometry in (or under) the typical cell size region, spherical fluorescent polystyrene beads were applied with 4.8 µm diameter and compared to beads having 15.8 µm diameter as reference.

Focusing criteria was studied and defined by Dino di Carlo considering the ratio of the bead diameter and channel size [15]:(4)$$\frac{a}{D_{h}}>0.07$$
where *a* is the bead diameter and *D_h_* is the hydraulic diameter of the channel:(5)$$D_{h}=\frac{2\;w\;h}{\left(w+h\right)}$$

The applied specific asymmetric channel geometry was designed with 50–100–150 µm critical width and 25–50–100 µm typical height—defined by the SU-8 thickness in process sequence—indicated by Figure 2. As summarized in Table 1. the lower channel height (defining low-aspect ratio (between 1–0.16) microchannels) is more advantageous considering the evolution of the focusing state in feasible flow rate ranges. In case of 100 µm channel height—which defines the aspect ratio between 2 and 0.66—, the lateral focusing of the smaller beads was predicted to be improbable.

The experiments confirmed the feasibility of these considerations. As shown in Figure 4, the lowest flow rate regime applicable to achieve focusing effect was experimentally determined for polystyrene beads having 4.8 µm and 15.8 µm diameter, respectively. The required flow rate values indicate extreme parameter dependency in the range, where *a*/*D_h_* is approaching the 0.07 critical value. In Figure 4 the continuous evolution of the focusing and size-dependent separation efficiency in the periodic structure is clearly shown, and size-dependent laterally separated focusing points appears at the outlet (after 23 periods) of the microfluidic system. Figure 5. demonstrates the behaviour of 4.8 µm- and 15.8 µm-sized particles in the channel with 25 µm height and 50 µm critical width (H25_W_cr_50).

According to the measurements, it was feasible to induce the focusing of the 4.8 µm beads at higher flow rates in channels having 50 µm channel height and 100/150 µm critical width (H50_W_cr_100/H50_W_cr_150) (see Table 1), although *a/D_h_* parameter is in the critical range in these cases. Accordingly, the channels with low aspect ratios (0.5/0.33) have advanced capabilities regarding the evolution of focusing states at higher flow rates. Our experiments have shown that effective focusing is observed for 15.8 µm beads due to the inertial lift forces acting more dominantly on larger beads, but in the case of too strict channel narrowing (at high aspect ratio), the effect of the Dean vortices, the Dean drag force become more dominant, and mixes the beads for H100_W_cr_50 channels. The local position and extent of focusing also can be deteriorated with the increase in flow rate. A similar disturbance visible in channel H50_W_cr_50 at 1 µL/s, although the focusing positions can be compressed with higher flow rates. However, single point lateral focusing was not possible for 4.8 µm beads, only a low level of lateral concentration was achieved, which shifted towards the channel mid center line at higher flow rates. The evolution of the Dean vortices in curving region of the curvilinear microfluidic channel in case different aspect ratios are presented in Figure 6.

The following question arises in heads, whether to increase the flow rate or increase the length of the channel. The answer depends on the tolerance of the sample to be separated what shear stresses it can withstand, which cannot be increased indefinitely. The other limitation is the length of the channel. In case the microfluidic system is to be integrated into an integral analytical device, dimensions must be chosen that are small enough to be compact but large enough to still perform its physical function, so the size is also limited. Below we look at some cases of what would happen if the restrictions could be ignored.

### 4.2. Flow-Rate-Dependent Lateral Focusing Efficiency

In case of the H50_W_cr_150 microchannel, the beads with 15.8 μm diameter, already focused on a flow rate of 1 μL/s, although *a/D_h_* parameter is less than 0.07, and the beads with 4.8 µm diameter were still focusable at 6 μL/s. Figure 7 represents the flow-rate-dependent positions of the investigated beads at the end of the curvilinear channel by experimental fluorescent images and Poincare maps calculated by Finite Element Modelling. Based on the experiments, the smaller beads started to concentrate at both edges of the channel at 3 µL/s. As the flow rate increased, their lateral position remained relatively stable, and at 6 μL/s, it began to focus already at one point laterally. According to COMSOL simulations, there is still a weak lateral concentration at 3 μL/s, but considering the cross-sectional Poincare maps, multiple focusing nodes can be distinguished. The 15.8 μm beads are vertically superimposed at two lateral points. The two bead sizes are concentrated laterally and almost overlap with an intensity maximum around the first trisect at the *y*-axis of the channel. Figure 7(B1) shows the overlap of the fluorescent signal of the 15.8 μm (purple) and 4.8 μm (white) beads. The focusing performances of the experimentally characterized 9 microfluidic structures are summarized and demonstrated in Figure A1 in Appendix A by presenting the fluorescent images recorded in case of different flow rates (0.5 µL/s, 1 µL/s, 3 µL/s) applied.

By further reduction of the critical cross-section (H50_W_cr_100), the smaller beads begin to focus relatively sooner occurring approximately at the flow rate of 3 µL/s. The maximum intensity of the beads is already visible in one focus and its position shifted laterally to the center of the channel contrary to H50_W_cr_150. The two bead sizes overlap in this case as well. By increasing the flow rate to 6 μL/s the two-bead size (15.8 μm and 4.8 µm) begin to separate, the smaller beads shift towards the inner wall of the channel.

The square-based constriction (H50_W_cr_50) had the least effective focusing, both streams of the beads overlap, and these are inseparable. At increased flow rate the focusing state of the larger beads deteriorated significantly, with intense mixing despite the higher *a/D_h_* parameter. Even in the case of FEM simulations, the lateral position is unstable at 6 µL/s. The simulation data are in good agreement with the experimental data.

It was proved by the results of the measurements and also FEM simulation, that lateral channel narrowing cannot improve the focusing efficiency at higher flow rates, where the Dean drag forces dominate the inertial effects.

### 4.3. Improved Lateral Focusing Efficiency in Channels with Low Aspect Ratio

In H50_W_cr_100 channels, although an appropriate lateral particle convergence was achieved, smaller 4.8 µm beads (green) were not completely focused and the 15.8 µm and 4.8 µm bead clusters could not be completely separated. At even lower channel height at the same critical width (H25_W_cr_100) and with higher flow rates (5–6 µL/s), enhanced lateral focusing was accomplished for each particle size, with proper bead cluster segregation as presented in Figure 8. With decreasing channel heights, the bead clusters lateral position shifted from the centerline to the bottom-side of the channel, while the bigger beads located closer to the center. Comparing the results with the simulations, they cover reality quite well. In case of H25_W_cr_100 channels at 5 µL/s flow rate (see Figure 8(C1–C3)) the smaller beads got more concentrated laterally and shifts towards the bottom side of the channel curvature, while the 15.8 µm beads moves towards the centerline.

The best separation results were obtained at 25 μm high channels as indicated by the experimental data of Figure 8. In microfluidic channels with H25_W_cr_150–H25_W_cr_100–H25_W_cr_50 parameters, the 4.8 μm beads were already focused at 0.5 μL/s. Despite the reduction of the critical width, the efficient lateral focusing of the larger beads also remained stable in these geometries. In these smaller cross-sections the beads reach their equilibrium position in a shorter time, but the extreme high speeds, and the secondary-flow-induced Dean drag forces could easily impair lateral focusing or spoil the bead cluster segregation. As proved experimentally and by in silico models also, the low aspect ratio channels have significant advantage regarding the focusing and separation efficiencies at these higher flow rates also.

### 4.4. Improving Lateral Focusing Efficiency by Multiple Periodes

As it was seen previously, in channels H50_W_cr_100 and H50_W_cr_150 the 4.8 µm beads could be focused only on extreme high flow rates. It makes sense that, by widening the channel, the smaller beads have to travel greater distance in the radial direction to reach their equilibrium position. Contrary to the lager particles, the same channel length—23 curvature in our case—is not sufficient to manage the lateral focusing of the 4.8 µm beads. In Figure 7 the focusing effect is visualized at channel H50_W_cr_150 by the experimental and simulation-based beads’ focusing tendency through the channel and the lateral distribution of the 15.8 µm and 4.8 µm beads at the 23rd curvature—at the end of the channel, at the flow rates of 0.5 and 3 µL/s. In H50_W_cr_150 at 0.5 µL/s none of the beads were focused laterally, and in simulations the 15.8 µm beads show a lateral bifurcation in the channel width (see Figure 7(A2)). The same lateral pattern was experimentally observed in case of 4.8 µm beads at 3 µL/s flow rate besides the laterally focused larger beads (see Figure 7(B1,B2)). In the case that the channel would be almost four times longer, the focusing and separation efficiency could be extremely improved at the same flow rate, at the end of the 100th curvature (see the simulation results in Figure 7(C2,C3)). By this solution, the focusing and separation could be manageable at lower shear rates—which could be beneficial for biological cells—although the maximal chip size should be considered.

## 5. Conclusions

In this work, the geometry-dependent inertial particle focusing and separation capabilities of periodically inhomogeneous curvilinear microfluidic channels were studied with special attention of smaller particles comparable to blood cells or bacteria. Both channel constrictions and high flow rates promote the evolution of Dean-induced drag forces which could deteriorate the clustering processes. By comparing different channel geometries, the advanced behaviour of channels with low aspect ratio regarding the particle focusing and separation efficiencies were proved experimentally and by FEM simulation also at higher flow rates. The beneficial effect of the increased number of periodic curves was also demonstrated by finite element simulation. Considering the smaller cells in real biological applications, these periodically inhomogeneous curvilinear microfluidic channels with low aspect ratio can be optimal solution for efficient size-dependent separation or focusing. The demonstrated microfluidic system can be capable to successfully manage the focusing and sorting living cells (as blood cells or tumor cells—being in the same size range as the rigid beads applied) although the specific size, shape (disc—red blood cells or rod—*E.coli* bacteria) and mechanical properties of the cells must be considered. The size- and shape-dependent movement of these cells in the flow field of the proposed microfluidic system is under investigations and will be discussed in a further study.

## Figures and Tables

**Figure 1 sensors-22-03474-f001:**
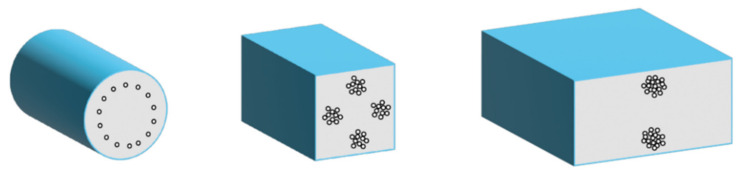
Particle focusing in microchannels with different cross-sectional geometry.

**Figure 2 sensors-22-03474-f002:**
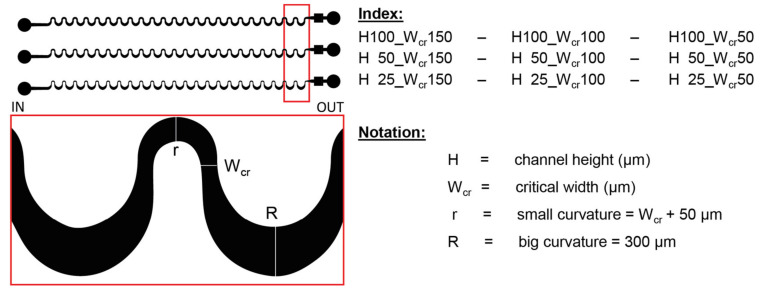
The schematic design of the lithographic mask representing the obtained geometrical parameters. Red box indicates the magnified area of the mask layout.

**Figure 3 sensors-22-03474-f003:**
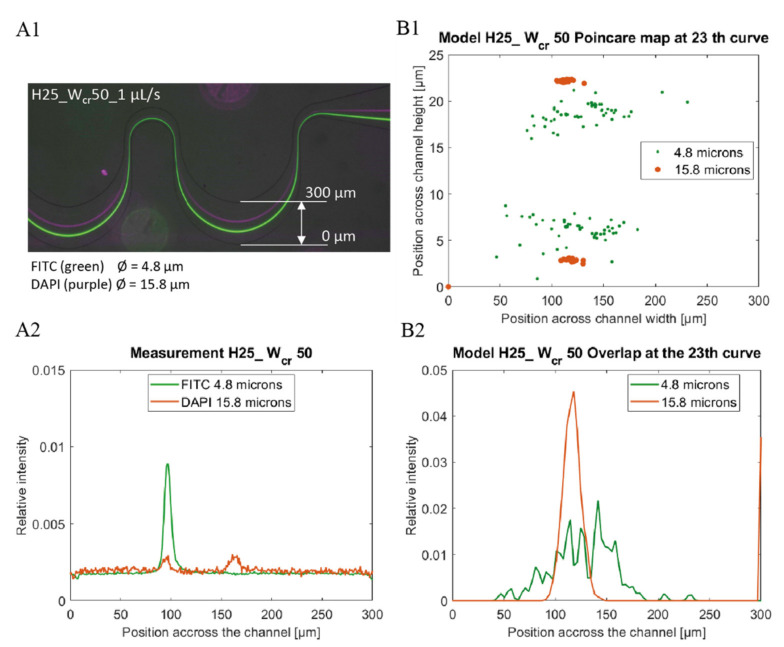
Comparison of Experimental (**A**) and Computational (**B**) data of fluorescent bead’s lateral focusing at 1 µL/s flow rate in channel H25_C50. The lateral distribution of 4.8 µm (green) and 15.8 µm (purple) beads are well separable in the 300 µm channel width (**A1**) and were represented based on the beads ’fluorescence intensity, and lateral position in channel width (**A2**). In the finite element simulation, the same bead distributions were examined in the channel cross section (**B1**), and in the channel width (**B2**) based on focusing efficiency.

**Figure 4 sensors-22-03474-f004:**
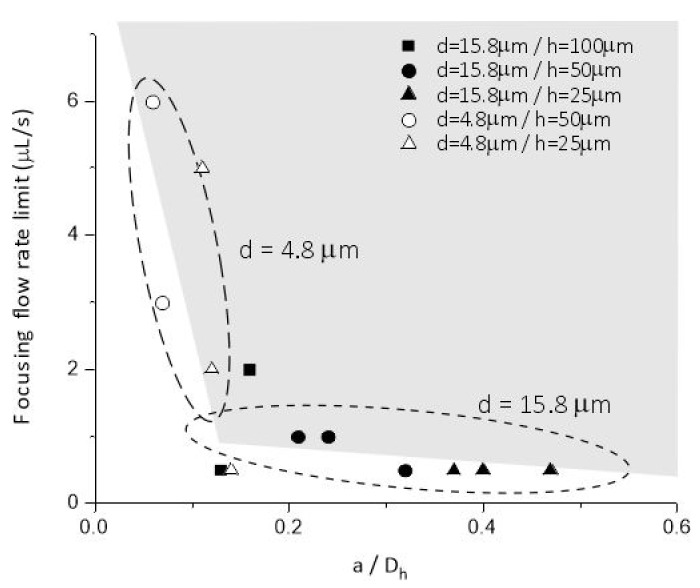
Minimal flow rates necessary for evolution particle focusing states as the function of the *a*/*D_h_* parameter, in case of 4.8 µm and 15.8 µm bead diameters as indicated by the dashed circles, respectively.

**Figure 5 sensors-22-03474-f005:**
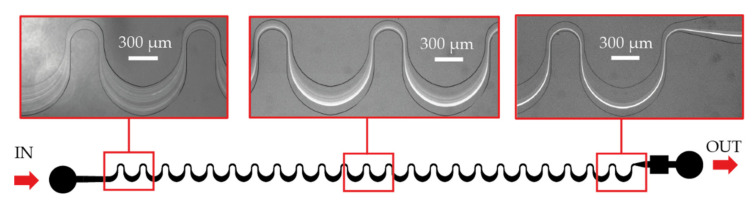
Evolution of the lateral focusing state of 4.8 µm fluorescent beads through the H25_W_cr_50 microfluidic channel from the beginning till the end (23rd curvature).

**Figure 6 sensors-22-03474-f006:**
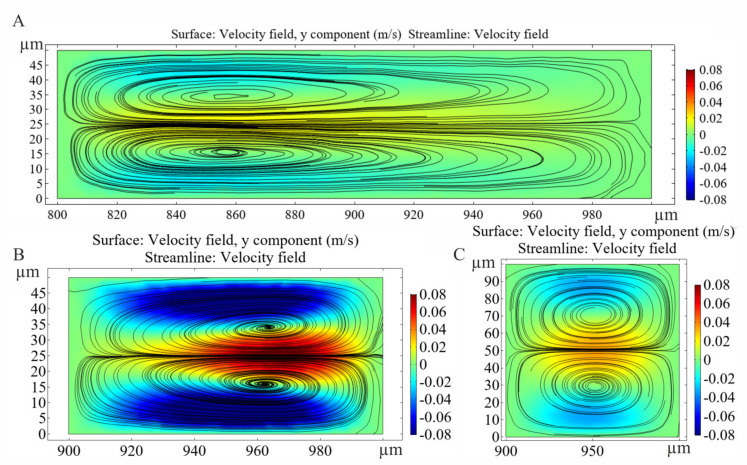
Evolution of Dean vortices in the H50_W_cr_150 (**A**), H50_W_cr_50 (**B**) and H100_W_cr_50 (**C**) microfluidic channels after the restrictions at 3 µL/s lateral flow rate. The y components of velocity field calculated by FEM.

**Figure 7 sensors-22-03474-f007:**
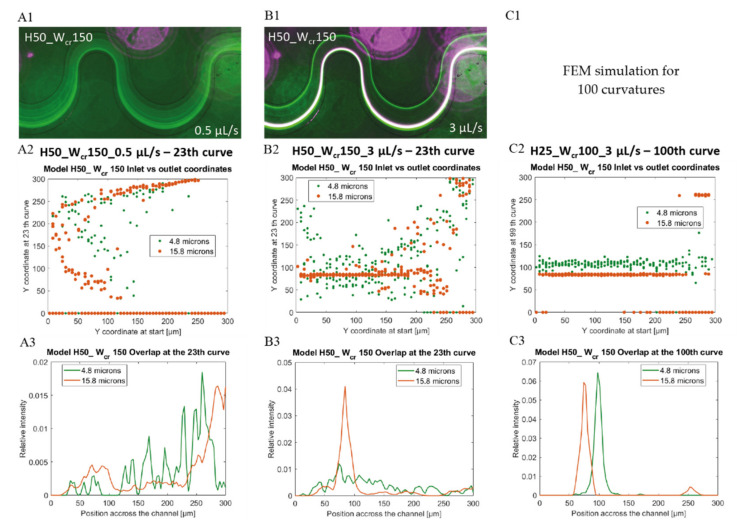
Lateral positions of 4.8 µm (green) and 15.8 µm (purple or orange) beads in microfluidic channel H50_W_cr_150 after 23 curves at flow rates 0.5 µL/s (**A1**–**A3**) and 3 µL/s (**B1**–**B3**). Experimental (**A1**,**B1** fluorescent images) and computational data of beads’ focusing tendency through the channel as *y* coordinates at the outlet vs. initial *y* coordinates (**A2**,**B2**,**C2**) and calculated lateral distribution at the end of the channel (**A3**,**B3**,**C3**). (**C1**) FEM simulation for 100 curvatures, (**C2**,**C3**) represents the modelled result after 100 curves in case of at 3 µL/s flow rate.

**Figure 8 sensors-22-03474-f008:**
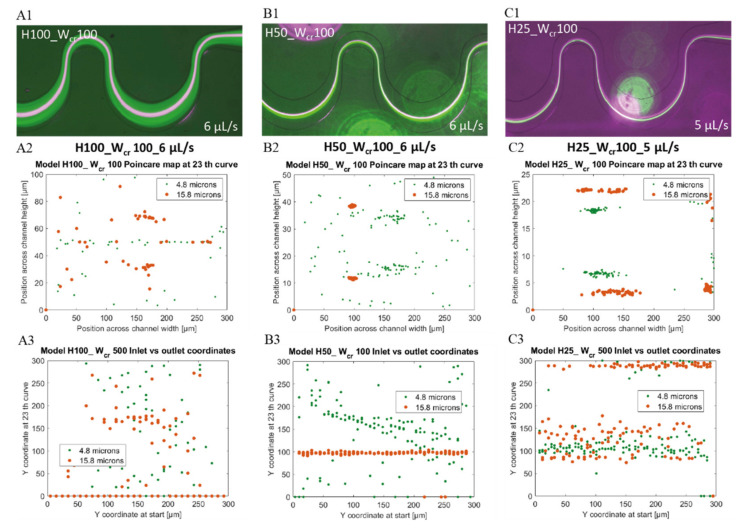
Position of 4.8 µm (green) and 15.8 µm (purple and orange) beads in microfluidic channels having 100 µm critical width and different channel heights of 100–50–25 µm at high flow rates (6 µL/s and 5 µL/s). Experimental data (**A1**,**B1**,**C1**) with 4.8 µm (green) and 15.8 µm (purple) beads—top view. Computational data of bead’s lateral distribution (Poincare map) at the end of the channel —cross sectional view (**A2**,**B2**,**C2**); and beads’ focusing tendency through the channel as y coordinates at the outlet vs. initial y coordinates (**A3**,**B3**,**C3**).

**Table 1 sensors-22-03474-t001:** Theoretical and experimental correlation on presence of lateral focusing effect. Shadow indicates the parameter sets which not complies the theoretical focusing criteria.

			Focusing Criteria (Calculation)		Minimal Flowratesfor Particle Focusing	
*H* [µm]	*W_cr_* [µm]	*D_h_* at *W_cr_*	15.8 µm*/D_h_*	4.8 µm*/D_h_*	*a =* 15.8 µm	*a =* 4.8 µm
100	150	120.00	0.13	0.04	0.5 µL/s	-
100	100	100.00	0.16	0.05	~2 µL/s	-
100	50	66.67	0.24	0.07	-	-
50	150	75.00	0.21	0.06	1 µL/s	6 µL/s
50	100	66.67	0.24	0.07	1 µL/s	~3 µL/s
50	50	50.00	0.32	0.10	0.5 µL/s	-
25	150	42.86	0.37	0.11	~0.5 µL/s	5 µL/s
25	100	40.00	0.40	0.12	~0.5 µL/s	2 µL/s
25	50	33.33	0.47	0.14	0.5 µL/s	0.5 µL/s

## Data Availability

Not applicable.

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
