# Peer review of "Geometry-Dependent Efficiency of Dean-Flow Affected Lateral Particle Focusing and Separation in Periodically Inhomogeneous Microfluidic Channels"

_sensors, 2022, doi:10.3390/s22093474_

Round 1

Reviewer 1 Report

This manuscript demonstrates that the meander-shaped asymmetric microchannel enables sheathless focusing and size-dependent sorting of microparticles. The authors showed that asymmetric inertial forces cause the lateral shift of the particles, which leads to the experimental separation of 4.8 μm and 15.8 μm beads. The mechanism of inertial focusing is supported by COMSOL FEM simulation by changing the parameters such as channel geometry and flow rates. The authors nicely conducted the fluorescent bead-based assay with newly designed microchannels, and discussed the results clearly. The reviewer believes that overall, this manuscript exhibits an exploratory study with highlighting technological microfluidic impacts. The reviewer recommends this manuscript for publication in Sensors without revision.

Author Response

Dear Reviewer,

Thank you very much for your kind effort, positive comments and supporting for publication our manuscript!

Yours sincerely,

Anita Bányai

Reviewer 2 Report

The article presented is well described and organized. The dean Flow affected particle focussing is well described and experimentally characterized. The reviewer has only 2 points to address, being a sensors journal and not specialized on  microfluidics a bit more on Dean flow in the introduction would help guide the general reader. The second point, although the sizes are in the range of bacteria as indicated along the text, bacteria have motility and might add additional instabilities to the flow. Not demanded for this article but testing with active particles could bring better insight on this... perhaps if bacteria are mentioned comments on this point should be added. 

Author Response

Dear Reviewer,

Thank you very much for your kind effort, improving comments and supporting for publication our manuscript! We have absolutely accepted your suggestions, and added a more careful and comprehensive description of the Dean flow evolving in curved channels in the introduction.

Moreover we are happy with Your interest about the applicability of these microfluidic structure for life science technology, due to this aspect is in focus regarding our further work.

We have already be able to successfully manage sorting red blood cells (being in the same size range as the smaller beads applied - Please see the attachment.) in the microfluidics, and there were implemented similar test also with GFP-E.coli bacterias. The beauty and difficulty of these measurements would be that we must consider specific size, shape and mechanical properties of living cells. Instead of solid spherical particles, the donut shape and chopstick shape morphology brought some uncertainties. We would like to present the results in details in a further article together with the simulation of movement of cells in this specific flow field. So we added a short section to highlight these focusing possibilities utilising the microfluidic environment, although we did not go into detail in this manuscript.

Thank you very much for Your constructive advices which will improve the manuscript!

Yours sincerely,

Anita Bányai

Reviewer 3 Report

This is a nice paper with many technical details about flow focusing. Do you think that the device works on actual live cells? Would it be possible to test on blood?

1. The authors develop a system of focusing of beads. 2. The research is relevant and detailed to the field of flow focusing. 3. Other people could use this system to focus beads. 4. It is interesting to consider whether the device can focus cells the same way as beads or how variations in cell size could be focused in a microfluidic device. 5. The conclusions are consistent. 6. References are appropriate and detailed. I would say this paper is ready for publication in present form.

Author Response

Dear Reviewer,

Thank you very much for your kind effort, improving comments and supporting for publication our manuscript! Moreover we are happy with Your interest about the applicability of these microfluidic structure for life science technology, due to this aspect is in focus regarding our further work.

We have already be able to successfully manage sorting red blood cells (being in the same size range as the smaller beads applied - Please see the attachment.) in the microfluidics, and there were implemented similar test also with GFP-E.coli bacterias. The beauty and difficulty of these measurements would be that we must consider specific size, shape and mechanical properties of living cells. Instead of solid spherical particles, the donut shape and chopstick shape morphology brought some uncertainties. We would like to present the results in details in a further article together with the simulation of movement of cells in this specific flow field. So we added a short section to highlight these focusing possibilities utilising the microfluidic environment, although we did not go into detail in this manuscript.

Thank you very much for Your constructive advices which will improve the manuscript!

Yours sincerely,

Anita Bányai
